# Chemical Characterization and Impact of Nipple Fruit (*Solanum mammosum*) on the Characteristics of *Lactobacillus acidophilus* LA K

Ricardo S. Aleman [1], Dany Avila [2], Allan Avila [2], Jack N. Losso [1], David Picha [3], Zhimin Xu [1] and Kayanush Aryana [1,*]

[1] School of Nutrition and Food Sciences, Louisiana State University Agricultural Center, Baton Rouge, LA 70808, USA; rsantosaleman@lsu.edu (R.S.A.); jlosso@agcenter.lsu.edu (J.N.L.); zxu@agcenter.lsu.edu (Z.X.)

[2] Faculty of Technological Sciences, Universidad Nacional de Agricultura Road to Dulce Nombre de Culmí, Km 215, Barrio El Espino, Catacamas 16201, Honduras; danijoelad@gmail.com (D.A.); xavieravila1994@gmail.com (A.A.)

[3] School of Plant, Environmental and Soil Sciences, Louisiana State University Agricultural Center, Baton Rouge, LA 70803, USA; dpicha@agcenter.lsu.edu

[*] Correspondence: karyana@agcenter.lsu.edu

**Abstract:** Nipple fruit (*Solanum mammosum*) has been considered to have great pharmaceutical potential because of its high amounts of solamargine and solasonine. This study aimed to examine the effect of nipple fruit at different concentrations (0.5%, 1%, and 2% ($w/v$)) on the viability, acid, bile, lysozyme, and gastric juice tolerance, and protease activity of *Lactobacillus acidophilus* LA K. The viability was studied in MRS broth. Acid tolerance was determined by adjusting the pH to 2, whereas bile tolerance was examined with oxgall 0.3% ($w/v$) in MRS broth. Lysozyme resistance was investigated in an electrolyte solution with lysozyme (100 mg/L), while gastric juice tolerance was analyzed with pepsin and NaCl. Protease activity was determined spectrophotometrically at 340 nm in skim milk with o-phthaldialdehyde reagent. *L. acidophilus* LA K was incubated anaerobically (37 °C). Microbial growth was determined every 2 h for 10 h of incubation. Acid tolerance was determined at 0, 5, and 15 min, whereas bile tolerance was analyzed at 0, 4, and 8 h of incubation. Lysozyme tolerance was determined at 0, 1, and 2 h of incubation, while gastric juice tolerance was determined at pH 2, 3, 4, 5, and 7. Protease activity was evaluated at 0, 12, and 24 h incubation. Nipple fruit's chemical and bioactive compounds were also examined to discuss their impact on the survival of *L. acidophilus* LA K. Nipple fruit did not affect microbial growth, bile, and acid tolerance. Nipple fruit at 2% had higher survivability on the simulated gastric juice and lysozyme resistance and increased protease activity.

**Keywords:** *Lactobacillus acidophilus*; nipple fruit; protease activity; lysozyme; gastric juices

## 1. Introduction

In recent years, there has been growing interest both from the scientific community and the general population regarding the role of *Lactobacillus acidophilus* in human health [1]. Among the mechanisms of action of the *Lactobacillus* strain, it produces antimicrobial substances such as lactic acid, hydrogen peroxide, diacetyl, and bacteriocins [2]. These compounds reduce the number of viable pathogen cells and affect bacterial metabolism or toxin production. The acids decrease intestinal pH, favoring the growth of beneficial microorganisms and increasing resistance to colonization by competing with pathogens to bind to adhesion sites on the gastrointestinal epithelium's surface. These substances also help to stimulate the immune response by promoting innate and acquired immunity, protecting against intestinal disease, encouraging the production of IgA (Immunoglobulin A), activating macrophages, and increasing the concentration of IFN-gamma (interferon

gamma). It has been suggested that *L. acidophilus* has beneficial effects in treating diarrhea, lactose intolerance, inflammatory bowel diseases, and colon cancer [3].

Nevertheless, these bacteria should be able to survive the transit through the intestinal system and proliferate within the gut. This implies that they should be capable of refilling in the presence of bile and gastric secretions or be absorbed alongside food material to allow longevity and transit through the intestine [4]. The International Dairy Federation believes that consuming approximately a product containing $10^6$–$10^7$ CFU (colony-forming units) per gram of viable bacteria is necessary to obtain therapeutic benefits.

As a result, new prebiotic sources are demanded, especially plant-based functional ingredients with complex carbohydrates, such as fiber and resistant starch [5]. *Solanum mammosum* (SM), a tropical fruit and a member of the *Solanaceae* family, has long been utilized for medical reasons. This fruit is known for its steroidal alkaloids like solasodine [6]. Nipple fruit (*Solanum mammosum*) is native to tropical South America (Antigua, Barbados, Belize, Bolivia, Colombia, Costa Rica, Cuba, Ecuador, Guadalupe, Jamaica, Guatemala, Honduras, Mexico, Nicaragua, Panama, Peru, Puerto Rico, Saint Lucia, and Saint Vincent), where it grows in lowland forests, grasslands, and degraded areas, mainly at low altitude [6,7]. The plant's phytochemicals included catechins, tannins, alkaloids, simple phenols, flavanones, cyanogenic glycosides, saponins, and triterpenes [7]. *Solanum mammosum* is also known for its steroidal glycoalkaloids, and the fruit has been reported to contain diosgenin and phytosterols [7]. The fruit's pulp is used for elephantiasis and to treat wounds, ulcers, and hemorrhoids for medicinal purposes. The leaves are used to treat skin conditions. The oil from the seeds is used for constipation and is applied as an anti-asthmatic and anti-inflammatory agent [8]. As a result, the present research seeks to examine the impact of chichigua on *L. acidophilus* LA K's attributes to examine its possible application in fermented products for human health purposes. It intends to evaluate the potential use of nipple fruit and probiotic bacteria for the development of new probiotic foods.

## 2. Materials and Methods

### 2.1. Plant Material

The nipple fruit was collected from Barrio Concepción, Marcala Municipality, La Paz Department (Honduras), between January and March 2022. A solution of nipple fruit (10% wt./wt.) was prepared and then kept cryogenically (−80 °C). The frozen solution was then freeze-dried with a (Liotop, São Carlos, SP, Brazil) lyophilizer for 48 h (−75 °C; 0.1–0.5 Pa). The freeze-dried material was grounded in a knife mill Retsch SM 100 (Retsch GmbH, Nordrhein-Westfalen, Germany) (501–700 mm). The freeze-dried nipple fruit powder was stored in sterile bags.

### 2.2. Experimental Design

The nipple fruit was examined for its ability to act as an antioxidant and its total phenolic, carotenoids, sugar, and organic acid content. The viability, acid, bile, lysozyme, and gastric juice tolerances and protease activity of *L. acidophilus* LA K (Danisco, Dairy Connection, Madison, WI, USA) as affected by nipple fruit powder were examined at 0% (control), 0.5%, 1%, and 2%. The bacterial viability was studied in MRS broth. Acid tolerance was determined by adjusting the pH to 2, whereas bile tolerance was examined with Oxgall 0.3% (*w/v*) in MRS broth. Lysozyme resistance was investigated in an electrolyte solution with lysozyme (100 mg/L), while gastric juice tolerance was analyzed with pepsin and NaCl. Protease activity was determined spectrophotometrically at 340 nm in skim milk with o-phthaldialdehyde reagent. Microbial growth was determined at 0, 2, 4, 6, 8, and 10 h of incubation. Acid tolerance was determined at 0, 5, and 15 min, whereas bile tolerance was analyzed at 0, 4, and 8 h of incubation. Lysozyme tolerance was determined at 0, 1, and 2 h of incubation, while gastric juice tolerance was determined at pH 2, 3, 4, 5, and 7. Protease activity was evaluated at 0, 12, and 24 h of incubation. *L. acidophilus* LA K was

incubated anaerobically (37 °C). The log counts were measured in MRS agar with duplicate readings. All experiments were performed in triplicate [9].

### 2.3. Proximate Analysis

The proximate *S. mammosum* (nipple fruit) analysis was carried out with the AACC and AOAC methods [10,11]. In the moisture (AACC method 44_01.01) analysis, moisture determination was based on the residue remaining after the heating samples (1 g) in aluminum pans at 105 °C for three hours in a drying oven (VWR International oven, Model 1370 GM, Sheldon Manufacturing Inc., Cornelius, OR, USA). In the ash (AACC method 08_01.01) analysis, ash refers to the content of minerals in the sample, such as calcium, phosphorus, potassium, magnesium, and iron. The determination was based on subjecting the sample to a temperature of 600 °C in a Thermolyne Type 600 muffle furnace (Thermo Scientific, Lawrence, KS, USA) to remove the organic matter destroyed at this temperature. In the fat analysis, the crude fat content in the rice flour was determined using the Soxhlet extraction method with Soxtec equipment (Soxtec System HT6, Tecator AB, Höganäs, Sweden) using hexane as a solvent at a temperature of 155 to 210 °C, with three grams of macerated sample under the following conditions: boiling for 60 min, rinsing for 10 min, solvent recovery for 10 min, drying for 10 min after extraction in the Soxtec, and oven drying at 105 °C for three hours. Residual fat was weighed on an analytical balance (Fisher scientific, ML-T). The protein analysis was carried out by the micro-Kjeldahl (Labconco Kansas City, MO, USA) technique with the AACC 46-13.01 method. In this analysis, 0.2 g of flour was weighed. The protein %, 5.95, was used as a conversion factor in the nipple flours. The total dietary fiber content of the nipple fruit was determined by the AACC enzyme gravimetric method (2001). The method was based on quantifying complex organic substances that are resistant to hydrolysis by digestive enzymes. It consisted of weighing 1 g of the dry and defatted sample (Soxhlet extraction), by quadrupling it in beakers, and then adding 50 mL of pH 6.0 phosphate buffer (the sample pH was corroborated to be 6.0) and 0.1 mL of thermostable amylase (Sigma A3306) to incubate for 15 min at 95 °C. The samples were shaken at 5-min intervals. The solutions were allowed to cool at room temperature. Then, 10 mL of NaOH 0.275 N was added, and the pH value was adjusted to 7.5. A protease solution (50 mg/mL) was prepared in phosphate buffer (pH 6.0), and 0.1 mL was added to each vessel. Subsequently, the samples were incubated with the protease for 30 min at 60 °C and with continuous agitation. The samples were cooled to room temperature, and after cooling, 10 mL of HCl was added to 0.325 M. The pH was at 4.5, and 0.1 mL of amyloglucosidase was added. Then, the samples were incubated for 30 min at 60 °C, with continuous agitation. Subsequently, 4 volumes of 95% ethanol were added to precipitate the soluble dietary fiber and left at rest at room temperature for 15 h. The solution was passed through a filter (40–60 microns) containing 0.5 g of Celatom (Sigma C8656). The Celatom was evenly redistributed within the filter using 78% ethanol, using suction with the vacuum pump. The suction was maintained to wash the sample. Washes were performed by adding 60 mL of 78% ethanol, 40 mL of 95% ethanol, and 40 mL of acetone. Of the four treated samples, two were used to determine the protein content by the Kjeldahl method and two were used for ash.

### 2.4. Total Polyphenols

The procedure was followed similarly to that of Swain and Hillis (1959) [12] with slight modification. 100 mg ($\pm$0.5 mg) of freeze-dried nipple fruit was added into 10 mL of 80% methanol, which was followed by vertexing (1 min), sonicating (15 min), and centrifuging (at 1230$\times$ *g* for 5 min). Total phenolics compounds (TPCs) were measured using Folin–Ciocalteu reagent and a colorimetric technique. In a 25 mL test tube, 0.5 mL of supernatant was added along with 8 mL of distilled water and 0.5 mL of Folin–Ciocalteu reagent. After 3 min, 1 mL of 1 N sodium bicarbonate ($Na_2CO_3$) was mixed, and the solution (0.5 mg/mL) was kept for 2 h at room temperature (25 °C). A Lambda 35 UV/Vis spectrophotometer (Perkin Elmer Instruments, Norwalk, CT, USA) was used to measure

the absorbance at 750 nm. The standard was gallic acid, and a gallic acid calibration curve (concentration range: 50–300 g/mL) was produced. The total phenolic content was calculated as mg gallic acid equivalent/g dry weight (mg GAE/g).

### 2.5. Radical-Scavenging Activity Assay

When an antioxidant reduces free radicals, their absorption at 517 nm vanishes. Using the Brand-Williams et al. (1995) method [13] with a few minor changes, the antioxidant potential was measured. Free radicals were generated using the compound DPPH (1, 1-diphenyl-2-picrylhydrazyl) [14]. Trolox (6-hydroxy-2, 5, 7, 8-tetramethylchroman-2-carboxylic acid) was utilized as the reference. Approximately 0.1 mL of the samples was dissolved with an additional 0.4 mL of 80% methanol and transferred to a 1.5 mL centrifuge tube, adding 0.5 mL fresh DPPH solution (0.01577 g/100 mL), which was prepared with 80% methanol. The obtained solution was mixed in the dark for 2 min before being dark-incubated at room temperature (25 °C) for 30 min. The percentage suppression of DPPH was determined from the reduction in absorption by using Equation (1):

$$\text{Antioxidant activity } (\%) = \left( \frac{\text{Absorbance control} - \text{Absorbance sample}}{\text{Absorbance control}} \right) \times 100 \quad (1)$$

A Lambda 35 UV/Vis spectrophotometer was used to detect the reduction in DPPH absorption at 517 nm (Perkin Elmer Instruments, Waltham, MA, USA). The 80% methanol solution without DPPH was used as the control. The antioxidant potential was determined using a Trolox calibration curve.

### 2.6. Total Carotenoids Content

Carotenoids were examined by Cano et al. 2019 [15]. Approximately 1 g of the freeze-dried material was combined with 0.5 g of magnesium carbonate in 50 µL of (all-E)-β-apo-8′-carotenal (0.40 mg/mL). The obtained solution was mixed with 20 mL of tetrahydrofuran solution (THF) with 0.1% ($w/v$) butylated hydroxytoluene (BHT). The mixture was homogenized in an Omnimixer (OMNI Macro S®, OMNI International, Kennesaw, GA, USA) at $3000\times g$ for 3 min. The mixture was ultrasonicated in a water bath (J.P. Selecta S.A., Barcelona, Spain) at 50/60 Hz and 360 W for 30 min. The solution was then centrifuged at $15,000\times g$ for 10 min at 4 °C, and the supernatant was collected. Twenty mL of acetone was mixed into the pellet, and the sample was extracted 3 times. The supernatant was mixed using 20 mL of diethyl ether at the end to reach a colorless pellet. The obtained solution was mixed with 20 mL of saturated water containing 30% ($w/v$) NaCl. The organic phase was recovered and dried for 10 min at room temperature with 2.5 g of anhydrous sodium sulfate. The sample was filtered using Whatman No. 1 filter paper, and the filtered sample was concentrated within a rotor evaporator at 30 °C. The extract was diluted to 2 mL of MeOH/MTBE/H$_2$O (45.5:52.5:2, $v/v/v$) and filtered through a 0.45 µm filter. A UV-Vis spectrophotometer measured the total carotenoid concentration at 450 nm (Specord 210 Plus Analytik, Jena, Germany). Hexane was used as a reference. The total carotenoid content (g carotenoids/g vegetable oil) was estimated using Equation (2).

$$\text{Total carotenoid content } \left( \frac{\mu g}{g} \right) = \frac{A \times V}{E1\%1cm \times P \times 100} \times 10^6 \quad (2)$$

where A is the absorption at 450 nm, V is the total volume of absorption (mL), E1%1cm is the absorption coefficient of the carotenoids mixed in hexane ((ξ = 2500 dL/g cm), and P is the amount of extract (g).

### 2.7. Organic Acid Profile

A 0.5% standard solution (Sigma Chemical Co., St. Louis, MO, USA) of individual organic acids (citric, tartaric, L-malic, quinic, and succinic) was utilized according to Zeppa et al. (2001) [16] with some modifications. The same HPLC model, the same column

(sugar HPLC model), and a Model 160 UV sensor with a specified frequency of 214 nm were utilized. The sensor was adjusted at 0.100 AU, and the calibration curve of the pulse was electrically combined in the external calibration mode by a Vista 401 integrator (Varan Assoc., Sunnyvale, CA, USA) with an attenuation of 4 and a chart speed of 0.5 cm/min. A 2-μ£ Rheodyne Model 7302 column inlet filter (Rheodyne, Cotati, CA, USA) and a 40 × 4.6 mm ion exclusion filter capsule filled with Aminex HPX-85H resin were inserted between the injector and column (Bio-Rad Laboratories, Hercules, CA, USA). Degassed 0.0008 N H2S04 fluid was made by dissolving chemical-level sulfuric acid concentration in HPLC-grade solution as the mobile phase at 0.8 mL/min flow velocity.

## 2.8. Determination of Sugar Profile

Approximately 100 mg of fructose, maltose, glucose, and sucrose (Sigma Chemical Co., St. Louis, MO, USA) were each dissolved in 80% ethanol, boiled, and filtered before being injected into the HPLC. The standards generated the calibration curve. To prepare the nipple fruit extracts, 10 g of nipple fruit powder was mixed and homogenized in 80% ethanol using a Vitris 45 homogenizer for 1 min. The obtained mixture was boiled for 15 min and filtered through a 0.2 μm pore size syringe filter (Whatman™ Puradisc, Buckinghamshire, UK). The samples were injected into a Beckman series 340 liquid chromatography system equipped with a pump (model 112), an injector (model 210) connected with a 20/IL sample loop, and an index refraction sensor (model 156). A 300 mm × 7.8 mm id. column coated with Aminex HPX-87C resin was used at 75 °C and a speed of 0.5 cm/min. A Varian 401 integrator electronically connected the detector signal. A 2~ Rhea- dyne 7302 column intake filter, a 40 × 4.6 mm ion exclusion keeper capsule equipped with Aminex HPX-85H resin, and a 40 × 4.6 mm anion/OH guard cartridge packed with Aminex A-25 resin were inserted between both the injector and the analytical column. The mobile phase was HPLC grade at a fluid rate of 1.2 mL/min [17].

## 2.9. Bacterial Viability

The microbial growth of *L. acidophilus* LA K was evaluated using the method of Aleman et al. (2023) [18]. The MRS broth containing nipple fruit powder (0% (control), 0.5%, 1%, and or 2%) and lactose (0.5% *w/v*) was autoclaved at 121 °C for 15 min. Aseptically, the cultured (1010 CFU/mL) (10% *w/v*) was added to the MRS broth containing the nipple fruit powder and lactose. The broth with the culture bacteria was incubated at 37 °C. The samples were plated for several time periods (0, 2, 4, 6, 8, and 10 h).

## 2.10. Bile Tolerance Test

The bile tolerance of *L. acidophilus* LA K was examined using the Pereira and Gibson (2002) method [19] with slight changes and some modifications. MRS broth (CriterionTM, Hardy Diagnostics, Santa Maria, CA, USA) containing the nipple fruit powder (0% (control), 0.5%, 1%, and or 2%), sodium thioglycolate (Sigma-Aldrich, St. Louis, MO, USA) (0.2% *w/v*), lactose (0.5% *w/v*), and oxgall salt (0.3% *w/v*) were autoclaved at 121 °C for 15 min. Aseptically, the cultured (10% *w/v*) was added to the MRS broth containing the nipple fruit powder, lactose, sodium thioglycolate, and oxgall salt. The broth with the culture bacteria was incubated at 37 °C. The samples were plated for several time periods (0, 4, and 8 h).

## 2.11. Lysozyme Resistance

The lysozyme resistance of *L. acidophilus* LA K was evaluated using the method of Zago et al. (2011) [20] with some slight modifications. The sterile electrolyte solution containing the nipple fruit powder (0% (control), 0.5%, 1%, and or 2%), 0.22 g/L $CaCl_2$, 6.2 g/L NaCl, 2.2 g/L KCl, and 1.2 g/L $NaHCO_3$ was used in the presence of filter sterilized lysozyme (100 mg/L). Aseptically, the cultured (10% *w/v*) was added to the electrolyte solution containing the nipple fruit powder, $CaCl_2$, NaCl, KCl, $NaHCO_3$, and lysozyme. The electrolyte solution with the culture bacteria was incubated at 37 °C. The samples were plated for several time periods (0, 1, and 2 h).

*2.12. Acid Tolerance and Gastric Juice Tolerance*

The acid tolerance of *L. acidophilus* LA K was evaluated using the Pereira and Gibson (2002) [19] method with slight modification. The MRS broth containing nipple fruit powder (0% (control), 0the .5%, 1%, and or 2%) and lactose (Sigma-Aldrich) (0.5% $w/v$) was adjusted to pH 2.0 with 1N HCl and autoclaved at 121 °C for 15 min. Aseptically, the cultured (10% $w/v$) was added to the MRS broth containing the nipple fruit powder and lactose. The broth with the culture bacteria was incubated at 37 °C. The samples were plated for several time periods (0, 5, and 15 min).

The gastric juice tolerance of *L. acidophilus* LA K was evaluated using the García-Ruiz et al. (2014) [21] and Liao et al. (2019) [22] methods with some minor modifications. The gastric juice containing the nipple fruit powder (0% (control), 0.5%, 1%, and 2%), filter sterilized pepsin (0.32% $w/v$) (Sigma-Aldrich, St. Louis, MO, USA), and NaCl (0.2% $w/v$) was adjusted to pH 2, 3, 4, 5, and 7 with 0.1 N HCl and NaOH and autoclaved at 121 °C for 15 min. The pepsin was filter sterilized. Aseptically, the cultured (10% $w/v$) was added to the gastric juice containing the nipple fruit powder, pepsin, and NaCl. The gastric juice with the culture bacteria was incubated at 37 °C. The samples were plated at 0 and 30 min of incubation to determine the viable bacteria.

*2.13. Protease Activity*

The protease activity of *L. acidophilus* LA K was evaluated using the Oberg et al. (1991) [23] method with some modifications. The skim milk containing the nipple fruit powder (0% (control), 0.5%, 1%, and 2%) was autoclaved at 121 °C for 15 min. The sterile skim milk (2.5 mL) was mixed with 1 mL of distilled water and 10 mL of 0.75 N trichloroacetic acid (TCA). The obtained mixture was filtered using Whatman Number 2 filter paper for 10 min. The TCA filtrate (3 mL) was mixed with OPA reagent (150 uL). The o-phthaldialdehyde solution was prepared by mixing 25 mL of 100 mM sodium borate (Fisher Scientific, Waltham, MA, USA) solution, 2.5 mL of 20% ($w/w$) SDS solution (Fisher Scientific, Waltham, MA, USA), 40 mg of o-phthaldialdehyde reagent (Alfa Aesar, Ward Hill, MA, USA) dissolved in 1 mL methanol (Sigma, St. Louis, MO, USA), and 100 μL of β-mercaptoethanol (Sigma, St. Louis, MO, USA) and diluting to 50 mL with distilled water. The absorbance was measured at 340 nm using a UV–Vis spectrophotometer (Nicolet Evolution 100, Thermo Scientific; Madison, WI, USA).

*2.14. Enumeration of L. acidophilus*

The MRS broth agar of *L. acidophilus* LA K was prepared with commercial MRS broth (Difco, Becton, Dickinson and Co., Sparks, MD, USA) supplemented with agar (1.5%) (Fisher Scientific, Fair Lawn, NJ, USA) in 1 L of distilled water. The MRS broth agar was adjusted to a pH of 5.2 using 1 N HCl. The media were boiled under stirring conditions and autoclaved at 121 °C for 15 min (Vargas et al., 2015) [24]. The samples were plated serially with 99 mL of sterilizing phosphate buffer 0.1% ($w/v$) solutions. Approximately 1 mL of the sample was placed in sterile Petri dish plates. The samples were anaerobically incubated at 37 °C for 72 h. A Quebec Darkfield Colony Counter (Leica Inc., Buffalo, NY, USA) was used to count the plates.

*2.15. Statistical Analysis*

Data were examined using PROC GLM (General Linear Model). The main effects (nipple fruit concentration and time) and interaction effect (nipple fruit concentration * time) were investigated using the least squares means. Significant differences were determined at $p < 0.05$, and the data were processed using Statistical Analysis Systems software (SAS 9.4).

## 3. Results and Discussion

*3.1. Chemical Constituents of Solanum Mammosum*

The *S. mammosum* (nipple fruit) proximate composition, sugar profile, organic acid compounds, antioxidant capacity, total phenolics, and total carotenoids are presented

in Table 1. Kuo (2002) [25] reported that *S. mammosum* fruit has 0.78 g/100 g of fiber, 3.23 g/100 g of carbohydrates, 0.96 g/100 g of protein, 0.05 mg/100 g of vitamin C, 4 mg/100 of β-carotene, and 7 mg/100 g of Ca. The low antioxidant activity of the fruit has been reported with an IC50 of 1706.95 μg micrograms (antioxidant capacity equivalent to Trolox) and a phenolic content of 3.08 g/100 g of crude extract of gallic acid [26].

**Table 1.** Nutritional composition of *Solanum mammosum* fruit.

| Parameters | |
| --- | --- |
| Moisture (g/100 g) | 93.76 ± 0.20 |
| Ash (g/100 g) | 0.56 ± 0.03 |
| Proteins * (g/100 g) | 1.07 ± 0.08 |
| Fat (g/100 g) | 0.07 ± 0.03 |
| Total dietary fiber | 0.77 ± 0.12 |
| Total carbohydrates ** (g/100 g) | 4.76 ± 0.45 |
| Citric acid (g/100 g) | 1.07 ± 0.13 |
| Tartaric acid (g/100 g) | 0.09 ± 0.01 |
| L- Malic acid (g/100 g) | 0.38 ± 0.05 |
| Quinic acid (g/100 g) | 0.48 ± 0.04 |
| Succinic acid (g/100 g) | 0.40 ± 0.06 |
| Glucose (g/100 g) | 0.40 ± 0.03 |
| Sucrose (g/100 g) | 3.84 ± 0.36 |
| Fructose (g/100 g) | 0.67 ± 0.04 |
| Total phenolics (μg GAE/mL) | 633.22 ± 10.67 |
| Total carotenoids (mg Q/mL) | 7.65 ± 0.32 |
| Antioxidant capacity (%) | 41.56 ± 2.77 |

The values are expressed as the mean ± standard deviation; each determination was made in triplicate; * (N × 6.25), ** calculated by difference: 100 − (moisture + protein + fat + ash + fiber).

The fruit reportedly has catechins, tannins, alkaloids, phenolic acids, flavonoids, saponins, steroids, and triterpenes [27]. The fruit has also been reported to contain diosgenin and phytosterols [28]. Other studies have reported other bioactive compounds, such as solanine, solanidine, and solasodine [29]. The solasodine in the fruits is around 0.2–1.2% by dry weight [30]. The solasodine content could increase in the green fruit until maturity (yellow fruit) and rapidly diminish as the yellow fruit progressed [29]. Solamargine induces hepatoma cell death (Hep3B) by apoptosis and involves the regulation of tumor necrosis factor I and II [31]. From the fruit, pseudoprotodioscin, protodioscin, and indioside D have been isolated [32]. Indioside D has been shown to possess antiproliferative activity as a panel of lines of human cancer cells [33]. The results suggest that indoside D induces apoptosis in HeLa cells via death's intrinsic and extrinsic pathways. The nipple fruit extract showed moderate activity in a study of 46 species with screening for antimalarial activity [34]. *S. mammosum* was shown to have compounds that could be medicinally valuable from a chemical point of view.

### 3.2. Bacterial Viability

The microbial growth of *L. acidophilus* LA K over 10 h of incubation after adding the nipple fruit is shown in Figure 1. The nipple fruit concentration effect and the interaction effect (nipple fruit concentration × time) were not significant ($p > 0.05$), whereas the time (hour) effect was significant ($p < 0.05$) (Table 2, Table 3). Nipple fruit did not affect the growth of *L. acidophilus* LA K, and the growth increased over time. Marcia et al. (2023) [9] reported that carao (*Cassia grandis*) did not adversely impact the viability of *L. acidophilus* LA K when they were added at 2% in MRS broth. The growth of *L. acidophilus* LA K in MRS broth increased the log counts in the first 3 h. Najim and Aryana (2011) [35] reported that *L. acidophilus* LA K exponential growth was reached in 4 h (MRS broth). Liong and Shah (2005) [36] observed that *L. acidophilus* LA K prevailed in the first 9 to 15 h after reaching the stationary phase when considering the cholesterol removal capacity. Thousands of bioactive compounds are widely found in exotic plants and fruits. The bioactivity and bioavailability

of these compounds can be transformed or metabolized by probiotics in the human gut. *L. acidophilus* LA K has been reported to metabolize dietary plant glucosides and externalize their bioactive phytochemicals [37]. Furthermore, *L. acidophilus* LA K can improve Chinese wolfberry juice's bioactive phytochemicals, antioxidant activities, and flavor profiles [38]. As a result, lactic acid bacteria fermentation (LAB) has been suggested to be an effective technology for prolonging shelf life and improving the flavor and aromatic and nutritional profile of foods [39,40]. Therefore, it is desired that functional ingredients with fortification purposes should not have significant inhibitory effects against LAB. Plants with significant phytochemical content may improve the growth characteristics of probiotics because of their bioactive compounds and may improve the gut microbiota [24].

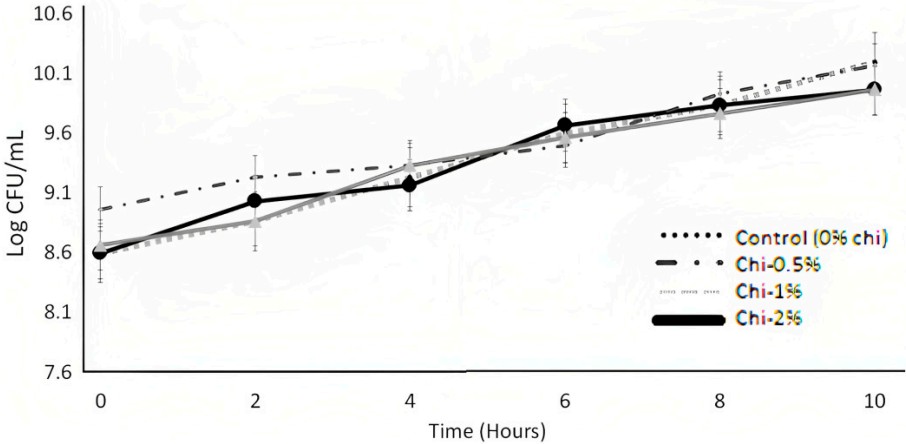

**Figure 1.** Viability (37 °C) of *L. acidophilus* LA K in MRS broth as influenced by the nipple fruit (Chi) concentration over 10 h. Average of three replicates. Error bars represent SE. Chi = *Solanum mammosum*.

**Table 2.** The *p*-value of nipple fruit (Chi) concentration, time, or pH, and their interaction for the bacterial viability, bile tolerance, acid tolerance, resistance to gastric juices, protease activity, and lysozyme resistance of *Lactobacillus acidophilus* LA-K.

| Effect | *L. acidophilus* **LA-K** |
|---|---|
| Viability | |
| Chi concentration | 0.183 |
| Time (Hours) | <0.0001 |
| Chi concentration × time | 0.2769 |
| Bile tolerance | |
| Chi concentration | 0.0893 |
| Time (Hours) | <0.0001 |
| Chi concentration × time | 0.3019 |
| Acid tolerance | |
| Chi concentration | 0.087 |
| Time (Minutes) | <0.0001 |
| Chi concentration × time | 0.2869 |
| Resistance to gastric juices | |
| Chi concentration | 0.0035 |
| pH | <0.0001 |
| Chi concentration × pH | 0.0218 |
| Protease activity | |
| Chi concentration | 0.0032 |
| Time (Hours) | <0.0001 |
| Chi concentration × time | 0.2396 |
| Lysozyme resistance | |
| Chi concentration | 0.0003 |
| Time (Minutes) | <0.0001 |
| Chi concentration × time | 0.4755 |

Chi = Solanum mammosum.

**Table 3.** Least squares means for the bacterial viability, bile tolerance, acid tolerance, resistance to gastric juices, protease activity, and lysozyme resistance of *Lactobacillus acidophilus* LA-K as influenced by the nipple fruit (Chi) concentration.

| Test | *L. acidophilus* LA-K |
|---|---|
| **Bacterial viability** | |
| Chi 0% (Control) | NS |
| Chi 0.5% | NS |
| Chi 1% | NS |
| Chi 2% | NS |
| **Bile tolerance** | |
| Chi 0% (Control) | NS |
| Chi 0.5% | NS |
| Chi 1% | NS |
| Chi 2% | NS |
| **Acid tolerance** | |
| Chi 0% (Control) | NS |
| Chi 0.5% | NS |
| Chi 1% | NS |
| Chi 2% | NS |
| **Resistance to gastric juices** | |
| Chi 0% (Control) | 8.34 [A] |
| Chi 0.5% | 8.30 [A] |
| Chi 1% | 8.55 [B] |
| Chi 2% | 8.57 [B] |
| **Protease activity** | |
| Chi 0% (Control) | 0.307 [A] |
| Chi 0.5% | 0.318 [A] |
| Chi 1% | 0.343 [A,B] |
| Chi 2% | 0.377 [B] |
| **Lysozyme resistance** | |
| Chi 0% (Control) | 6.01 [A] |
| Chi 0.5% | 6.12 [A] |
| Chi 1% | 6.31 [B] |
| Chi 2% | 6.79 [C] |

[A,B,C] Means within the same column alongside the same test with different letters differ statistically ($p < 0.05$). NS = not significant differences among the control and chichigua samples. Chi = *Solanum mammosum*.

*3.3. Bile Tolerance*

The bile tolerance of *L. acidophilus* LA K over 8 h of incubation after adding the nipple fruit is shown in Figure 2. The nipple fruit concentration effect and the interaction effect (nipple fruit concentration × time) were not significant ($p > 0.05$), whereas the time (hour) effect was significant ($p < 0.05$) (Table 2, Table 3). The nipple fruit did not affect the bile tolerance of *L. acidophilus* LA K, and the growth decreased over time. The growth remained stable until 4 h and significantly ($p < 0.05$) decreased at 8 h. Other studies have reported similar growth for *L. acidophilus* LA K inoculated in MRS broth [41,42]. Vargas et al. (2015) [43] reported the bile tolerance of L. bulgaricus over 8 h of incubation in MRS broth, with it having 0.3% oxgall. Whey protein increased the log counts in the broth with 0.3% oxgall compared with the broth without whey protein. Commonly, the bacterial cell membrane is favorably vulnerable to bile salts, which can damage the cell membrane, inducing the leakage of dangerous intracellular material [24]. Tolerance to bile salts is considered an essential criterion in selecting dietary ingredients applied to probiotic bacteria. Although a degree of tolerance is required for optimum growth in the intestine, it is reasonable to select the functional ingredients that could make more resistant microorganisms since this allows its use in lower doses, thus guaranteeing the product's safety and endogenous microbiota balance [44]. For this reason, it is desired that functional ingredients with fortification purposes should at least not have significant inhibitory effects against LAB in digestive enzymatic activities.

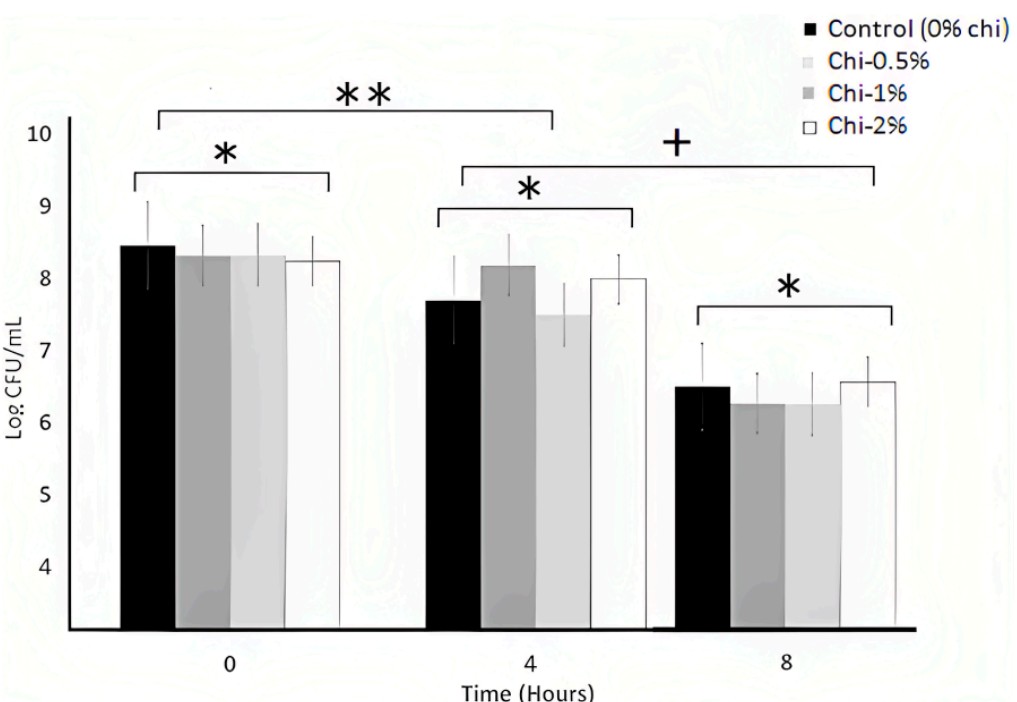

**Figure 2.** Bile tolerance (0.3% oxgalt) of *L. acidophilus* LA K in MRS broth as influenced by the nipple fruit (Chi) concentration over 8 h. * Average of three replicates. Values with different letters are significantly different (*p* < 0.05). Error bars represent SE. Chi = *Solanum mammosum.* indicates non-significant differences (*p* > 0.05) among the treatments of 0, 4, and 8 h. ** indicates non-significant differences (*p* > 0.05) among the treatments of 0 and 4 h. + Indicates significant differences (*p* < 0.05) among the hours of 4 and 8.

Some mechanisms of action of plant extracts inhibit the activity of digestive enzymes to protect probiotic bacteria from gastrointestinal stress [42]. *S. mammosum* extract inhibits enzymatic activities such as AChE, α-and β carboxyl, AcP, and AkP at 80 mg/mL [45]. Phytochemical compounds have been shown to inhibit digestive enzyme activities as well. Many previous studies have shown that plants with high polyphenol content have good lipase inhibitory effects [46]. Camila et al. (2017) [47] reported that tannin-rich extracts exhibited potent enzymatic inhibition on porcine pancreatic α-amylase, and Batiha et al. (2020) [48] showed that licorice has anti-pepsin activity. In our study, chichigua has a polyphenol content of only 3.07%, and it did not affect the bile tolerance and lipid solubilization of *L. acidophilus* LA K cell membranes at 0.5%, 1%, and 2%.

### 3.4. Acid Tolerance and Resistance to Gastrointestinal Juices

Acid tolerance was studied at pH 2 to examine the effect of the nipple fruit on the log counts of *L. acidophilus* LA K, mimicking stomach conditions. The acid tolerance of *L. acidophilus* LA K over 15 min of incubation as affected by the nipple fruit concentration is shown in Figure 3. The nipple fruit concentration effect and the interaction effect (nipple fruit concentration × time) were not significant (*p* > 0.05), whereas the time (minutes) effect was significant (*p* < 0.05) (Table 2, Table 3). The nipple fruit did not affect the acid tolerance of *L. acidophilus* LA K, and the growth decreased over time. The growth significantly (*p* < 0.05) decreased at 0 to 10 min and remained stable from 10 to 15 min. The buffering ability of possible compounds contained in the nipple fruit was inadequate to prevent bacteria decay under acidified conditions. Our results indicate that the enzymes in the cell walls of *L. acidophilus* LA K were not influenced at all by the nipple fruit components.

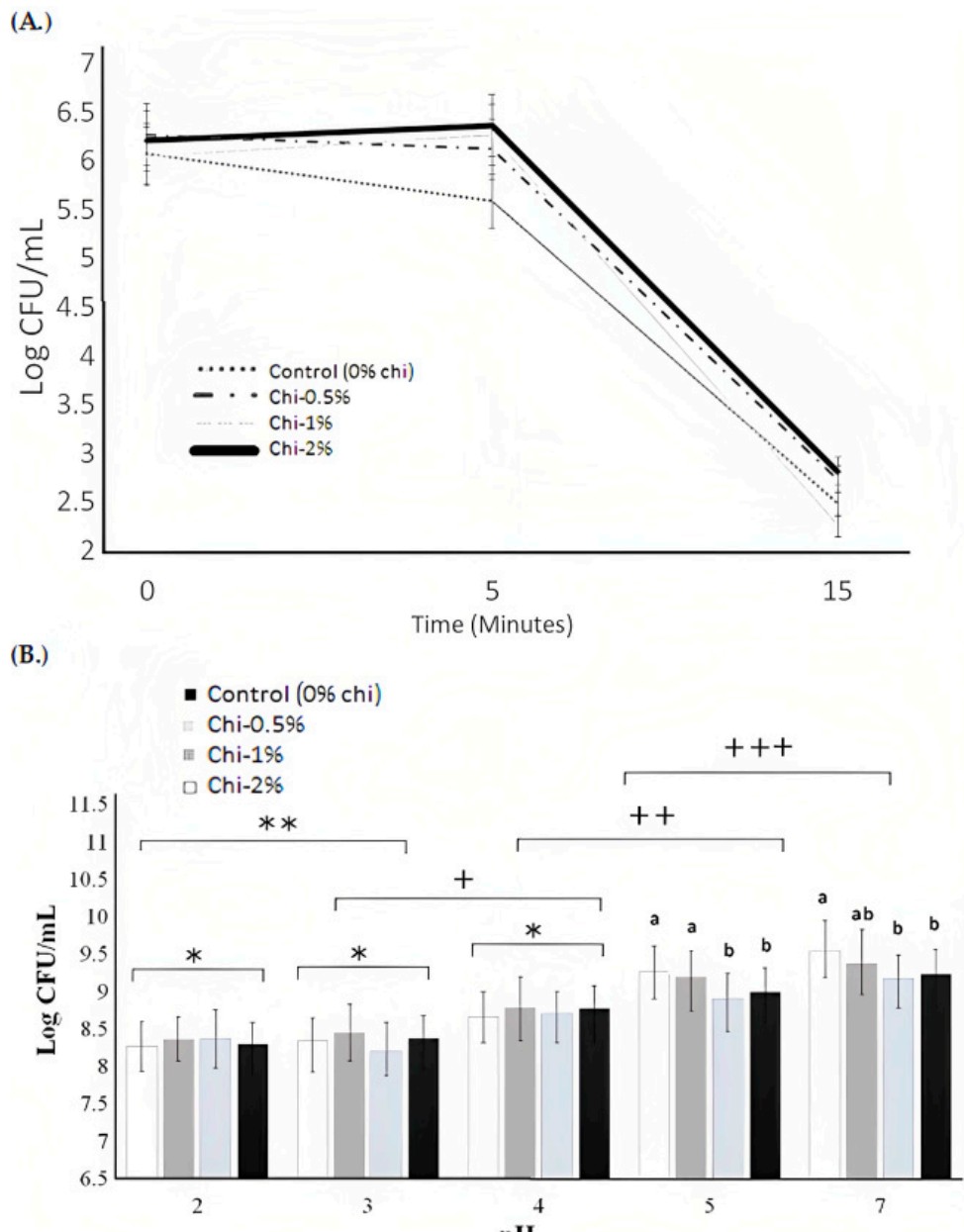

**Figure 3.** (**A**) Acid tolerance (pH 2) of *L. acidophilus* LA K in MRS broth as influenced by the nipple fruit (Chi) concentration over 15 min. average of three replicates. Error bars represent SE. Chi = *Solanum mammosum*. (**B**) Resistance of *L. acidophilus* LA K to simulated gastric juice (pH 2, 3, 4, 5, and 7) in formulated gastric juice solution as influenced by the nipple fruit (Chi) concentration over different pH conditions. Chi = *Solanum mammosum*. [a,b] Means with different letters mean significant differences ($p > 0.05$) among the treatments at pH 5 and 7. * indicates non-significant differences ($p > 0.05$) among the treatments at pH 2, 3, and 4. ** indicates non-significant differences ($p > 0.05$) at pH 2 and 3. + indicates significant differences ($p < 0.05$) at pH 3 and 4. ++ indicates significant differences ($p < 0.05$) at pH 3 and 4. +++ indicates significant differences ($p < 0.05$) at pH 5 and 7.

Acid tolerance is considered one of the main properties used to select potential bacteria as probiotics [49]. The acidic pH reduces microbial growth by increasing the hydrogen ion concentration [H+], which reduces their ability to produce ATP, proteins, and vital nutrients for their development [24]. Before reaching the intestinal tract, probiotics must survive to transit through the stomach, where gastric acid secretion constitutes a defense mechanism against ingested microorganisms [50]. The energy for this process comes from

forming an electrochemical gradient composed of charged ions (H+) provided by the electron transport chain that pumps H+ into the opposite compartment to that of the F1 complex that synthesizes ATP [51]. *L. acidophilus* LA K is not tolerant to acid environments (pH 2) [42]. Nevertheless, Lactobacillus acidophilus can be more acid resistant to other probiotics such as *Lactobacillus casei* and *Bifidobacterium bifidum* [36,50]. Probiotics must survive the acidic stress from the digestive system to colonize the small intestine, and ingredients with fortification must, at least, not influence the probiotics' survivability to the acidic gastric system.

The gastrointestinal system is a physiological barrier in the different parts of the digestive system (the esophagus, small intestine, and large intestine) that probiotics encounter, conforming to different pH values with the action of proteolytic enzymes such as pepsin, NaCl, and bile salts [52]. The resistance of *L. acidophilus* LA K to gastric juice tolerance at different pH values is shown in Figure 3. The gastric juice tolerance was analyzed at different pH values (2, 3, 4, 5, and 7) for bacterial survivability to examine the effect of nipple fruit concentration on the log counts of *L. acidophilus* LA K. The interaction effect (nipple fruit concentration and pH), pH effect, and nipple fruit concentration effect were significant ($p > 0.05$) (Tables 2 and 3), meaning that the control and nipple fruit samples did not follow the same trend. At pH 5 and 7, higher log counts were found in the nipple fruit samples (1% and 2%) compared to the control samples. As the pH when down, the log counts were lower. These results are consistent with other studies showing that Lactobacillus strains remain viable when exposed to pH values of 2.5–4.0 but exhibit a loss of viability at lower pH values [16]. Kim et al. (2008) [53] reported that *L. acidophilus* LA K was completely destroyed after 1 h of incubation at pH 1.2, and Favaro-Trindale and Grosso (2002) [54] showed that no strains of *L. acidophilus* (La-05) survived in the artificial gastric environment of pH 1.0 after 1 h. Nipple fruit could improve the log counts in gastric juices of *L. acidophilus* LA K in pH 5 and 7, as nipple fruit has considerable amounts of carbohydrates and sucrose (data shown in Table 1). Sucrose has been demonstrated to improve the growth of lactic acid bacteria [55]. Mituoka (1992) [56] reported that *L. acidophilus* LA K is most active in the small intestine, where the pH slowly rises from pH 6 to about pH 7.4. The results indicate the potential of nipple fruit to improve the resistance of *L. acidophilus* LA K in the small intestine, where the pH ranges from pH 6 to about pH 7.4.

*3.5. Resistance to Lysozyme*

The mouth is the first barrier that probiotics undergo through the gastrointestinal tract. As an essential part of the non-specific immune defense mechanism, lysozyme is a crucial component of antibacterials in saliva. It participates in the host's non-immune defense against bacteria, maintaining the stable balance of the oral cavity environment [57]. The biological activity of lysozyme causes interference with bacterial growth, especially in Gram (+) bacteria, hydrolyzing 1,4-beta-bonds between N-acetylmuramic acid and N-acetylglucosamine in the bacterial membrane of the bacteria [57]. Resistance to lysozyme is shown in Figure 4. The nipple fruit concentration and time effects were significant ($p > 0.05$), whereas the interaction effect (nipple fruit concentration $\times$ time) was not significant ($p < 0.05$) (Tables 2 and 3). The interaction effect was not significant ($p > 0.05$), meaning that the control and nipple fruit samples followed the same trend. The control and nipple fruit samples decreased in the log counts over time. For the control and nipple fruit treatments, the log counts decreased from 0 to 60 min and remained stable from 60 to 120 min. The 1% and 2% nipple fruit electrolyte dispersions reported significantly ($p > 0.05$) higher viability than the control samples (Table 2). Nipple fruit has considerable amounts of sucrose (data shown in Table 1), and this substrate could be used for the survivability of *L. acidophilus* LA K. Lysozyme degrades peptidoglycan, an essential component of the prokaryotic cell walls of prokaryotes outside the cytoplasmic membrane [58]. This polymeric structure is essential for bacterial cells since it carries out the maintenance of their morphology, serves as a scaffold to fix other components in the cell envelope, such as proteins, deals with mechanical stress, and allows nutrients to reach the plasma membrane

that contains the transporters (i.e., proteins, porins, and permeases, among others) necessary for their translocation in the cytosol, all processes necessary for the survival of the bacteria [58]. As a hypothesis, nipple fruit could inhibit the lysozyme enzymatic activity carried out by lysozyme, which hydrolyzes, attacks, and disrupts the glycosidic bonds of peptidoglycans. However, the action mechanisms of lysozyme inhibition by nipple fruit should be investigated thoroughly, and more examinations should be addressed for future research.

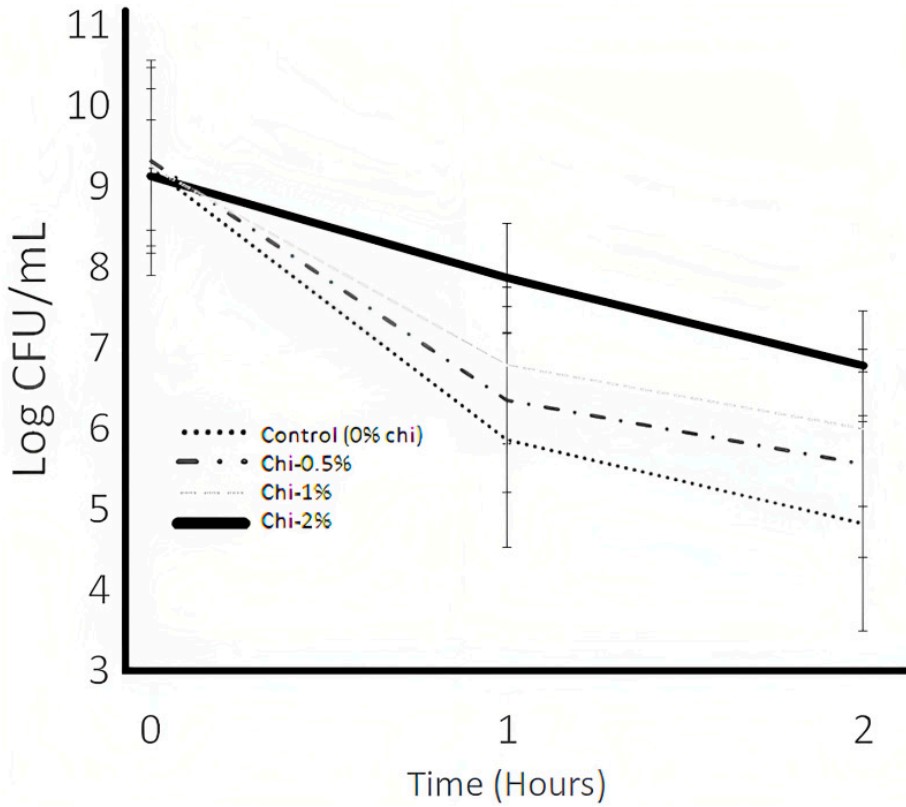

**Figure 4.** Resistance to lysozyme (100 mg/L) of *L. acidophilus* LA K in electrolyte solution as influenced by the nipple fruit (Chi) during the incubation time of 2 h. Average of three replicates. Error bars represent SE. Chi = *Solanum*.

### 3.6. Protease Activity

Proteolysis is the degradation of proteins by the action of the proteolytic system of lactic acid bacteria, which synthesizes peptidases such as aminopeptidases and dipeptidases [59]. The protease activity of *L. acidophilus* LA K is shown in Figure 5. The nipple fruit concentration and time effects were significant ($p > 0.05$), whereas the interaction effect (nipple fruit concentration × time) was not significant ($p < 0.05$) (Tables 2 and 3). The interaction effect was not significant ($p > 0.05$), meaning that the control and nipple fruit samples followed the same trend. The control and nipple fruit samples increased in the log counts over time. The protease activity of *L. acidophilus* LA K showed an increase after 24 h for the control and nipple fruit treatments (Figure 5). The nipple fruit treatments showed no significant difference ($p > 0.05$) from the control samples at 0 h and 12 h, whereas 1% and 2% nipple fruit had significantly higher protease activity at 24 h. Nipple fruit has significant amounts of sucrose, which could increase the cell density of *L. acidophilus* LA K, leading to higher protease activity. Lactic acid bacteria's development ability depends on their proteolytic system, allowing them to release essential amino acids for growth [60]. *L. acidophilus* NCFM synthesizes serine proteinase [61]. Proteolytic activity is vital for bacterial survival, and nipple fruit incorporation into skim milk could be suggested to enhance the growth of *L. acidophilus* LA K.

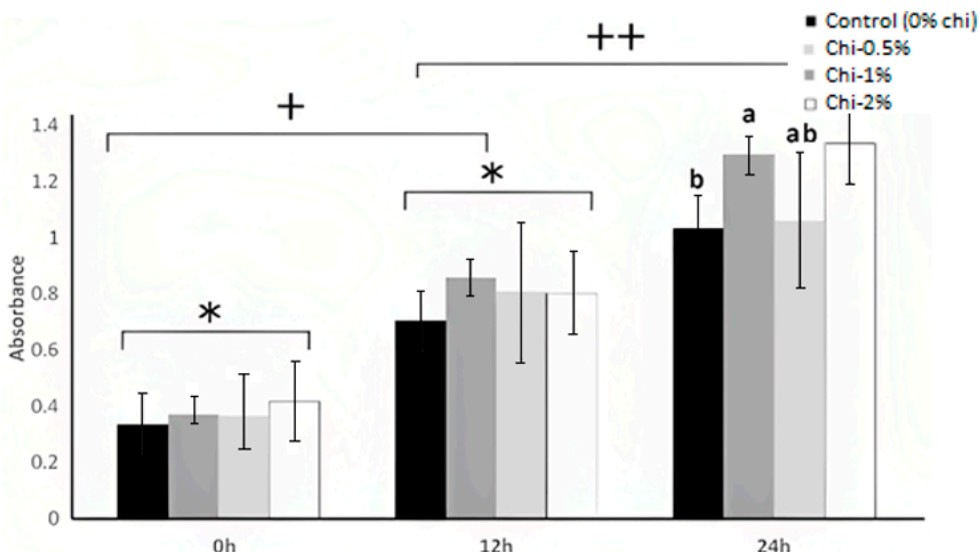

**Figure 5.** The protease activity (37 °C) of *L. acidophilus* LA K in skim milk as influenced by the nipple fruit (Chi) concentration over an incubation period of 24 h. Average of three replicates. Values with different letters are significantly different ($p < 0.05$). Error bars represent SE. Chi = *Solanum mammosum*. [a,b] Means with different letters mean significant differences ($p > 0.05$) among the treatments in 8 h. * indicates non-significant differences ($p > 0.05$) among the treatments in 0 and 4 h. [+] indicates significant differences ($p < 0.05$) among the treatments in 0 and 4 h. [++] indicates significant differences ($p < 0.05$) among the treatments in 4 and 8 h.

## 4. Conclusions

The results show that nipple fruit does not affect the bacterial viability and acid and bile tolerance of *L. acidophilus* LA K. Furthermore, adding 1% and/or 2% nipple fruit powder had higher log counts in gastric juices at pH 5 and 7. Similarly, 1% and/or 2% nipple fruit powder improved the lysozyme resistance and protease activity of *L. acidophilus* LA K. Overall, nipple fruit had no adverse effects on *L. acidophilus* LA K's characteristics. Consequently, nipple fruit at 1 and/or 2% may enhance the properties of *L. acidophilus* LA K in fermented dairy products. Nevertheless, more studies, such as on the probiotic characteristics of nipple fruit in in vivo models, are encouraged to allow for its specific application in prebiotic or symbiotic systems. In general, nipple fruit had some favorable impacts on probiotic properties, and it is encouraged to study nipple fruit in cultured products to determine its potential.

**Author Contributions:** Conceptualization, R.S.A. and K.A.; methodology, R.S.A., A.A. and D.A.; software, R.S.A.; formal analysis, R.S.A. (most of the research), A.A. and D.A.; resources, R.S.A., K.A. and D.P.; data curation, R.S.A.; writing—original draft preparation, R.S.A., Z.X. and J.N.L.; writing—review and editing, R.S.A., Z.X., J.N.L. and K.A.; project administration, R.S.A. and K.A.; funding acquisition, R.S.A. and K.A. All authors have read and agreed to the published version of the manuscript.

**Funding:** This research was funded by the University National of Agriculture (Honduras) (Ref. C-DSIP-008-2023-UNAG) and USDA Hatch funds LAB94511.

**Institutional Review Board Statement:** Not applicable.

**Informed Consent Statement:** Not applicable.

**Data Availability Statement:** Not applicable.

**Acknowledgments:** We wish to thank the Food Sciences, Louisiana State University Agricultural Center, and the Faculty of Technological Sciences, Universidad Nacional de Agricultura Road to Dulce, Catacamas, Olancho, Honduras.

**Conflicts of Interest:** The authors declare no conflict of interest.

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
