# Peer review of "Chemical Characterization and Impact of Nipple Fruit (Solanum mammosum) on the Characteristics of Lactobacillus acidophilus LA K"

_fermentation, doi:10.3390/fermentation9080715_

Round 1

Reviewer 1 Report

fermentation-2505985-peer-review-v1

The paper is interesting and contributes a new observation for the use of an exotic fruit as source of prebiotics for bacterial growth and survival. In my opinion paper can be accepted, however, some adjustments, clarification and additional information need to be considered by authors.

Abstract need to be presented better with focus on the observed results and not on the applied methods.

Ln33: Probiotic properties are strain specific, in this regards link between species and probiotic properties are a bit exaggerated. Please, correct the sentence.

Ln48: Please, present this numbers in appropriate way, with exponential position.

Ln52-60: Please, provide appropriate reference for this statement.

Please, for all experiments mentioned in the 2.2. (Ln78) provide appropriate references.

Ln81: In this and other occasions, after introducing the bacterial species, please, use the abbreviated way of writing the species name.

Ln82: After state name, please, add USA. Check for similar adjustments rest of the manuscript.

Ln96-102: Maybe some more details for the applied methods can be provided.

Ln110: In this and similar occasions, write the formula with numbers as indexes.

Ln112: Please, add USA after CT.

Ln131: After specific equipment was introduced, in second occasions, only name of the company need to be provided without full address.

Ln138: Please, in this and other occasions, standardize wt/v or w/v, but do not mix both way of expression.

Ln168: if I am not wrong, this is first time to mention Bio-Rad. If this is correct, please, provide the full address for the company.

Ln187-193; 195-200; 203-210: Please, explain better this section. Provide more details.

Ln212-219: Formulas need to be correct and indexed positions respected. Some additional details need to be added. 

Ln189: Chichigua powder? Is this correct?

Ln231-239: Some additional details for this method need to be supplied.

Ln241-243: Please, do not need to give these details. Just stated that commercial MRS broth was supplemented with 1.5% agar.

Ln258: Please, check the sentence. Looks like something is missing.

Ln276: Please, keep same way to express yourself - nipple or teat fruit.

Please, for the bacterial species, check and use new names suggested from April 2020.

Discussion can be presented a bit better. In current way, discussion is a bit superficial.

In conclusion section authors will need to be more objective.

References need additional attention and adjustments according to the instructions for authors.

Better quality of the figures needs to be provided.

Author Response

Reviewer 1

“Ln33: Probiotic properties are strain specific, in this regards link between species and probiotic properties are a bit exaggerated. Please, correct the sentence”

Response: the sentence is now corrected.

“Ln48: Please, present this numbers in appropriate way, with exponential position.”

Response: numbers are now corrected with exponential position

“Ln52-60: Please, provide appropriate reference for this statement.”

Response: references are now placed.

“Please, for all experiments mentioned in the 2.2. (Ln78) provide appropriate references.”

Response: references are now placed.

“Ln81: In this and other occasions, after introducing the bacterial species, please, use the abbreviated way of writing the species name.”

Response: abbreviation is now placed.

“Ln82: After state name, please, add USA. Check for similar adjustments rest of the manuscript.”

Response: USA is now placed.

“Ln96-102: Maybe some more details for the applied methods can be provided.”

Response: details for the applied methods are now added.

“Ln110: In this and similar occasions, write the formula with numbers as indexes.”

Response: formula with numbers as indexes are now included.

“Ln112: Please, add USA after CT.”

Response: USA is now placed.

“Ln131: After specific equipment was introduced, in second occasions, only name of the company need to be provided without full address.”

Response: the company is now provided without full address.

“Ln138: Please, in this and other occasions, standardize wt/v or w/v, but do not mix both way of expression.”

Response: w/v is now standardized though out the manuscript.

“Ln168: if I am not wrong, this is first time to mention Bio-Rad. If this is correct, please, provide the full address for the company.”

Response: full address for the company is now added.

“Ln187-193; 195-200; 203-210: Please, explain better this section. Provide more details.”

Response: All of these sections are linked with the experimental design section where more details are now provided.

“Ln212-219: Formulas need to be correct and indexed positions respected. Some additional details need to be added.” 

Response: formula with numbers as indexes were put these sections are linked with the experimental design section where more details are now provided.

“Ln189: Chichigua powder? Is this correct?”

Response: Chichigua powder is changed to nipple fruit powder.

“Ln231-239: Some additional details for this method need to be supplied.”

Response: Some additional details are now provided.

“Ln241-243: Please, do not need to give these details. Just stated that commercial MRS broth was supplemented with 1.5% agar.”

Response: this suggestion is now addressed.

“Ln258: Please, check the sentence. Looks like something is missing.”

Response: The sentence is rephased.

“Ln276: Please, keep same way to express yourself - nipple or teat fruit.”

Response: The nipple fruit is now used all the way.

“Please, for the bacterial species, check and use new names suggested from April 2020.

Discussion can be presented a bit better. In current way, discussion is a bit superficial.

In conclusion section authors will need to be more objective.

References need additional attention and adjustments according to the instructions for authors.

Better quality of the figures needs to be provided.”

Response: the discussion, conclusion, and figures are now improved.

Reviewer 2 Report

What was the letter-number strain designation of the Lactobacillus acidophilus used in the research? This letter-number designation should be in the title, abstract, conclusion, title of the figures, and in the text of the Manuscript. It should be clarified whether Lb. acidophilus used was probiotic (look at the line 401). Proper identification of the bacteria used as a probiotic is very important because the properties, and functional characteristics of bacteria are strain-dependent and cannot be extrapolated to species or genus.

line 33: it cannot be written: “this probiotic” without specifying (letter, and number) the strain.In abstract and in the aim of the work: it should be explain:  nipple fruit – chichigua.

Line: 435-436 : This conclusion cannot be drawn – the research weren’t on dairy products .

Line: 289-290: What was the matrix in these research (Najin and Aryana (2011))? – add this information to the text.

Figure 1: Add the information about the medium (MRS broth, 0,5% lactose) to the title of the Figure 1.

Figure 3: The title should contain the information about the pH value in the acid tolerance test.

Line 187. What was the inoculum [log CFU/ml] before being added to the MRS broth? Add this information to the p. 2.9.

Tolerance to simulated gastric juice was evaluated after 30 minutes of incubation. Why was the time so short? Food is longer digested in the stomach.

Some minor comments in the text of Manuscript.

Author Response

Reviewer 2

“What was the letter-number strain designation of the Lactobacillus acidophilus used in the research? This letter-number designation should be in the title, abstract, conclusion, title of the figures, and in the text of the Manuscript. It should be clarified whether Lb. acidophilus used was probiotic (look at the line 401). Proper identification of the bacteria used as a probiotic is very important because the properties, and functional characteristics of bacteria are strain-dependent and cannot be extrapolated to species or genus.”

Response: Lactobacillus acidophilus LA K is now mentioned as indicated.

“line 33: it cannot be written: “this probiotic” without specifying (letter, and number) the strain”

Response: Lactobacillus acidophilus LA K is now mentioned as indicated.

“In abstract and in the aim of the work: it should be explain:  nipple fruit – chichigua.”

Response: chichigua is changed to nipple fruit.

“Line: 435-436 : This conclusion cannot be drawn – the research weren’t on dairy products.”

Response: The sentence is now rephrased to “Thus, nipple fruit at 2% concentration may pottentially enhance the probiotic characteristics of S. thermophilus and L. bulgaricus in cultured dairy by-products. Nevertheless, it is encouraged to study nipple fruit in cultured products to determine its potential.

“Line: 289-290: What was the matrix in these research (Najin and Aryana (2011))? – add this information to the text.”

Response: the medium (MRS broth) is now added to the text.

“Figure 1: Add the information about the medium (MRS broth, 0,5% lactose) to the title of the Figure 1.”

Response: the information is now provided in the title of figure.

“Figure 3: The title should contain the information about the pH value in the acid tolerance test.”

Response: the information is now provided in the title of figure.

“Line 187. What was the inoculum [log CFU/ml] before being added to the MRS broth? Add this information to the p. 2.9.”

Response: the inoculum was at 10% v/v and it is now added to the experimental section.

“Tolerance to simulated gastric juice was evaluated after 30 minutes of incubation. Why was the time so short? Food is longer digested in the stomach.”

Response: The simulated gastric juice methodology was evaluated according to Marcia et al., (2023).

Reviewer 3 Report

The manuscript describes a study that is interesting when assessing the usefulness of a tropical fruit as food ingredients, combined with probiotic bacteria. The study includes two aspects. The first, relative to the study of physicochemical characteristics of the fruit, is well done. The second, aimed at seeing the impact of fruit lyophilized in some characteristics of a strain of L. acidophilus, has several problems, which are detailed below, in addition to other formal type questions.

- The authors must indicate in the title and throughout the text, when talking about the strain, the identification of the same (LAK). If only genus and species is written, it is indicating that all the strains of that genus and species have the same behavior, which is not true

  - Always use italics to write the genus and species of microorganisms. Check references. Also apply for "Solanaceae", Line 53.

-The manuscript abounds in many details but it is not clear the last objective of the study, which should appear clearly in the abstract and the introduction Is it intended to use fruit dust and probiotic bacteria for the development of new probiotic foods? (Liquids, solids ...?)

  There are many methodological inaccuracies:

. Acid tolerance: If you are chosen pH = 2 to simulate gastric acidity, "Gastric acid tolerance" should be titled. Or is pH 2 chosen for another reason? In what medium did the assay performed? In any case, it should be indicated that the strain has very low acidity tolerance .

. In another assay we talk about "Gastric juice tolerance" in a medium that simulates stomach juice but no details of its quantitative composition are given. The components are only mentioned. And among the pHs tested the value of 2 appears again. I do not understand again: when I speak of gastric juice, its pH is 2 but in the test other pH values are tested. Then there is talk about tolerance during the gastrointestinal transit and I can assume that this is why different phs were chosen but, then, the name of the assay cannot say "Gastric juice tolerance". Clarify everything  

- lines 346-353: unnecessary. Delete

- line 357: "L. acidophilus". in italics

- 3.5: change title: "Resistance to gastrointestinal juices" and join with 3.4. Rinse well all the item of tolerance to acid and gastrointestinal juices

- line 367: it is not Figure 4 but Figure 6

- line 369: it is not understood what the counts correspond to. Are counts measured after a certain time? How much time? What was the original count. Please clarify

- line 422: it is not Figure 6 but Figure 4

- Figure 1: why do you start from such high counts to see development? Normally, when this subject is investigated, one starts from a lower count, far from the stationary phase of growth.

Title: After "Viability" place in parentheses the environmental conditions of the assay (temperature, medium). Indicate the identification of the strain

  - Figure 2: after "Bile tolerance" indicate the concentration of bile used. Indicate the identification of the strain

- Figures 3 and 6: restate the whole subject and results of tolerance to acidity and make a single Figure, always indicating the identification of the strain and the environmental conditions of the assay

- Figure 4: after "activity" indicate in brackets the medium and temperature of the assay

-Figure 5: after "lisozime" indicate between brackets the concentration tested. Indicate the identification of the strain

- Improve Conclusions according to everything observed, clearly indicating if the objectives of the work were met and if the results have possible applications

Author Response

Reviewer 3

“The manuscript describes a study that is interesting when assessing the usefulness of a tropical fruit as food ingredients, combined with probiotic bacteria. The study includes two aspects. The first, relative to the study of physicochemical characteristics of the fruit, is well done. The second, aimed at seeing the impact of fruit lyophilized in some characteristics of a strain of L. acidophilus, has several problems, which are detailed below, in addition to other formal type questions.”

“The authors must indicate in the title and throughout the text, when talking about the strain, the identification of the same (LAK). If only genus and species is written, it is indicating that all the strains of that genus and species have the same behavior, which is not true”

Response: Lactobacillus acidophilus LA K is now mentioned in tittle and abstract as indicated.

  “Always use italics to write the genus and species of microorganisms. Check references. Also apply for "Solanaceae", Line 53.”

Response: the genus and species of microorganisms are italicized.  “Solanaceae” is also italicized.

“-The manuscript abounds in many details but it is not clear the last objective of the study, which should appear clearly in the abstract and the introduction Is it intended to use fruit dust and probiotic bacteria for the development of new probiotic foods? (Liquids, solids ...?)”

Response: objective of the study is now clearly stated in the introduction.

“ There are many methodological inaccuracies:

. Acid tolerance: If you are chosen pH = 2 to simulate gastric acidity, "Gastric acid tolerance" should be titled. Or is pH 2 chosen for another reason? In what medium did the assay performed? In any case, it should be indicated that the strain has very low acidity tolerance.”

Response: Acid tolerance is the name for one of the criteria for probiotic potential. This assay is to simulate the stomach acid conditions (Marcia et al., 2023).

“In another assay we talk about "Gastric juice tolerance" in a medium that simulates stomach juice but no details of its quantitative composition are given. The components are only mentioned. And among the pHs tested the value of 2 appears again. I do not understand again: when I speak of gastric juice, its pH is 2 but in the test other pH values are tested. Then there is talk about tolerance during the gastrointestinal transit and I can assume that this is why different phs were chosen but, then, the name of the assay cannot say "Gastric juice tolerance". Clarify everything “

Response: Acid tolerance is to simulate the stomach conditions and gastric juice tolerance simulated its full digestions. Two analyses were done to associate the results and have more robust results.  We followed the methods of Marcia et al., (2023).

“lines 346-353: unnecessary. Delete”

Response: lines 346-353 are now deleted.

“ line 357: "L. acidophilus". in italics”

Response: L. acidophilus". is italized

“3.5: change title: "Resistance to gastrointestinal juices" and join with 3.4. Rinse well all the item of tolerance to acid and gastrointestinal juices”

Response: acid tolerance and resistance to gastrointestinal juices are now combined.

“line 367: it is not Figure 4 but Figure 6”

Response: Figures are now combined as Figures 3A and B.

“ line 369: it is not understood what the counts correspond to. Are counts measured after a certain time? How much time? What was the original count. Please clarify”

Response: this sentence is now clarified.

“ line 422: it is not Figure 6 but Figure 4”

Response: Figures number order is now changed.

“Figure 1: why do you start from such high counts to see development? Normally, when this subject is investigated, one starts from a lower count, far from the stationary phase of growth.”

Response: We were just wanting to make sure the culture did not go through autolysis and could survive at high concentrations needed for the probiotic benefit. 

“Title: After "Viability" place in parentheses the environmental conditions of the assay (temperature, medium). Indicate the identification of the strain”

Response: temperature, medium and strain are now placed in Figure title / legend.

  “Figure 2: after "Bile tolerance" indicate the concentration of bile used. Indicate the identification of the strain”

Response: concentration of bile used and strain are now mentioned.

“Figures 3 and 6: restate the whole subject and results of tolerance to acidity and make a single Figure, always indicating the identification of the strain and the environmental conditions of the assay”

Response: Figures 3 and 6 are now combined. The identification of the strain and the environmental conditions are now included in figure legend.

“Figure 4: after "activity" indicate in brackets the medium and temperature of the assay”

Response: the medium and temperature are now indicated in figure legend.

“Figure 5: after "lisozime" indicate between brackets the concentration tested. Indicate the identification of the strain”

Response: concentration used and strain are now indicated in Lysozyme figure legend.

“Improve Conclusions according to everything observed, clearly indicating if the objectives of the work were met and if the results have possible applications”

Response: conclusions are now improved.

Round 2

Reviewer 2 Report

Line 260: it should be: „nipple”

Line 334: delete: lactic (double: ‘lactic”)

Line 391: L. acidophilus should be in italic

Line 470-471: add strain designation to the Lb. acidophilus.

Line 474-475: delete the sentence: “Thus, nipple fruit at 2% concentration may pottentially enhance the probiotic characteristics of S. thermophilus and L. bulgaricus in cultured dairy by-products.”

Point. 2.9.   MRS broth was inoculated with 10% (v/v) of the culture; I suggest add the information about cell concentration in inoculum (cfu/mL).

Author Response

- I return on the issue of "Acid tolerance": this, written, can indicate that the Acid tolerance will be studied in front of any acid. If you want to indicate that it is a gastric acid, it should be written: "Gastric Acid Tolerance." As for the "Gastric Acid Tolerance" is then written to indicate that gastrointestinal juices will be studied, it should be clearly indicated what the authors want to do, ergo  "gastrointestinal juices tolerance." If I write "gastric" I am referring to the stomach. Please , correct

Response: The terms were corrected.

Line 260: it should be: „nipple”

Response: the word nipple was corrected.

Line 334: delete: lactic (double: ‘lactic”)

Response: lactic was deleted.

Line 391: L. acidophilus should be in italic

Response: L. acidophilus was italiced

Line 470-471: add strain designation to the Lb. Acidophilus.

Response: The strain was added.

Line 474-475: delete the sentence: “Thus, nipple fruit at 2% concentration may pottentially enhance the probiotic characteristics of S. thermophilus and L. bulgaricus in cultured dairy by-products.”

Response: The sentence was deleted.

Reviewer 3 Report

The authors have significantly improved the manuscript. There are still some things to adjust according to the indications of the first review:

The identification of the strain used should be indicated throughout the text, not only in title and abstract. It should always be mentioned that strain is talking

- Use italics for genus and species. Missing References 6, 54 and 61

- Do not use italics for "bifidobacteria" (Ref. 19) and "lactobacilli" (Ref. 36). It is not Latin. Latin is used for genus and species

- Ref. 35: acidophilus, delbrueckii subp. bulgaricus

- Line 708 (Fig. 4): "solution"

-Conclusions, lines 475-6: "S. thermophilus and L. bulgaricus"?

- I return on the issue of "Acid tolerance": this, written, can indicate that the Acid tolerance will be studied in front of any acid. If you want to indicate that it is a gastric acid, it should be written: "Gastric Acid Tolerance." As for the "Gastric Acid Tolerance" is then written to indicate that gastrointestinal juices will be studied, it should be clearly indicated what the authors want to do, ergo  "gastrointestinal juices tolerance." If I write "gastric" I am referring to the stomach. Please , correct

Author Response

The authors have significantly improved the manuscript. There are still some things to adjust according to the indications of the first review:

- The identification of the strain used should be indicated throughout the text, not only in title and abstract. It should always be mentioned that strain is talking

Response: L. acidophilus LA K was put thought out the manuscript.

- Use italics for genus and species. Missing References 6, 54 and 61

Response:  references were corrected.

- Do not use italics for "bifidobacteria" (Ref. 19) and "lactobacilli" (Ref. 36). It is not Latin. Latin is used for genus and species

Response: references were corrected

- Ref. 35: acidophilus, delbrueckii subp. Bulgaricus

Response: references were corrected

- Line 708 (Fig. 4): "solution"

Response: the word solution was corrected

-Conclusions, lines 475-6: "S. thermophilus and L. bulgaricus"?

Response: the sentence was deleted.